# Group I Metabotropic Glutamate Receptors Modulate Motility and Enteric Neural Activity in the Mouse Colon

**DOI:** 10.3390/biom13010139

**Published:** 2023-01-09

**Authors:** Anita J. L. Leembruggen, Yuqing Lu, Haozhe Wang, Volkan Uzungil, Thibault Renoir, Anthony J. Hannan, Lincon A. Stamp, Marlene M. Hao, Joel C. Bornstein

**Affiliations:** 1Department of Anatomy and Physiology, School of Biomedical Sciences, University of Melbourne, Parkville, VIC 3010, Australia; 2Department of Immunology & Pathology, Monash University, Melbourne, VIC 3800, Australia; 3Florey Institute of Neuroscience and Mental Health, Parkville, VIC 3010, Australia

**Keywords:** enteric nervous system, gastrointestinal tract, metabotropic glutamate receptors, enteric glia, calcium imaging, colonic motor complexes

## Abstract

Glutamate is the major excitatory neurotransmitter in the central nervous system, and there is evidence that Group-I metabotropic glutamate receptors (mGlu1 and mGlu5) have established roles in excitatory neurotransmission and synaptic plasticity. While glutamate is abundantly present in the gut, it plays a smaller role in neurotransmission in the enteric nervous system. In this study, we examined the roles of Group-I mGlu receptors in gastrointestinal function. We investigated the expression of *Grm1* (mGlu1) and *Grm5* (mGlu5) in the mouse myenteric plexus using RNAscope in situ hybridization. Live calcium imaging and motility analysis were performed on ex vivo preparations of the mouse colon. mGlu5 was found to play a role in excitatory enteric neurotransmission, as electrically-evoked calcium transients were sensitive to the mGlu5 antagonist MPEP. However, inhibition of mGlu5 activity did not affect colonic motor complexes (CMCs). Instead, inhibition of mGlu1 using BAY 36-7620 reduced CMC frequency but did not affect enteric neurotransmission. These data highlight complex roles for Group-I mGlu receptors in myenteric neuron activity and colonic function.

## 1. Introduction

Glutamate is a non-essential amino acid that is abundantly available in the gut lumen. It is present in many dietary proteins, making up approximately 10% of amino acids in the average human diet [1,2]. In addition, it is synthesized by many different bacteria of the gut microbiome [3]. Glutamate is absorbed by intestinal epithelial cells and mostly metabolized, either for energy or converted to other molecules [4]. Therefore, while glutamate levels in the gut lumen can reach millimolars in concentration after a meal, the amount of free glutamate in systemic blood is relatively low [5]. 

In the central nervous system (CNS), glutamate is a major excitatory neurotransmitter. Glutamate acts through two main classes of ionotropic glutamate receptors, *N*-methyl-D-aspartate (NMDA) receptors and α-amino-3-hydroxy-5-methyl-4-isoxazolepropionic acid (AMPA) receptors, as well as via metabotropic glutamate receptors (mGlu) receptors. There are eight types of mGlu receptors that can be grouped according to their expression and function [6,7]. Group-I mGlu receptors comprise mGlu1 and mGlu5, which are expressed post-synaptically in the CNS [8,9] and are coupled to Gq proteins, whose activation leads to the formation of inositol 1,4,5-trisphosphate (IP3) and diacylglycerol [10,11]. Group-I mGlu receptors have an important role in neuronal plasticity, and their activation has been shown to facilitate both long-term potentiation and long-term depression at central synapses [12,13].

The enteric nervous system (ENS) is a network of neurons and glial cells located within the wall of the gut that is responsible for the control of many aspects of gastrointestinal function, including motility and secretion [14,15]. The ENS consists of many different subtypes of neurons, including sensory neurons, interneurons, and motor neurons which communicate using a variety of different neurotransmitters. In contrast to the CNS, acetylcholine is the major excitatory neurotransmitter in the ENS. Glutamate and its receptors are present in many different subtypes of enteric neurons [16,17,18,19]. Historically, most neurochemical and functional studies of the ENS have been carried out in guinea pigs, including the identification of NMDA and AMPA receptors [20,21]. Group-I mGlu receptors have been identified on submucous and myenteric neurons in the guinea pig ENS [22,23]. Interestingly, mGlu5 has also been found to be expressed in enteric glial cells in both guinea pigs and rats [16]. Functionally, both ionotropic and metabotropic glutamate receptors have been shown to mediate excitatory postsynaptic potentials in the guinea pig [19,24] and, more recently, mouse ENS [25]. However, the role of Group-I mGlu receptors in gut motility has not been previously investigated. 

In this study, we examined the distinct roles of Group-I mGlu receptors on enteric neurotransmission and colonic motility using live calcium imaging and ex vivo colonic motility recordings. Using RNAscope in situ hybridization, we examined the presence of mGlu1 and mGlu5 transcripts in the mouse myenteric plexus. In addition, we investigated whether there were changes in the composition of specific subtypes of enteric neurons in *mGlu5-knockout (mGlu5-KO)* mice. Our data suggest that Group-I mGlu receptors are involved in myenteric neuronal activity and have roles in the initiation of colonic motility; however, mGlu5-KO mice showed no clear changes in their ENS structure.

## 2. Materials and Methods

### 2.1. Animals

For ex vivo analysis of colonic motility and RNAscope in situ hybridization, male and female mice (C57BL/6 background, aged 8–14 weeks) were used. For calcium imaging experiments, *Wnt1-Cre;GCaMP6f* mice were used, where the fluorescent calcium indicator GCaMP6f is expressed in all enteric neurons and glia. *Wnt1-Cre;GCaMP6f* mice were generated by crossing *Wnt1-Cre* males (RRID:MGI:2386570 [26]) with *R26R-RCL-GCaMP6f* females (Ai95, The Jackson Laboratory, Stock# 028865; RRID:IMSR_JAX:028865 [27]). Mice were housed in the Biomedical Science Animal Facility at the University of Melbourne and were killed via cervical dislocation, in accordance with the University of Melbourne Animal Ethics Committee (Ethics ID: 1914927, accepted in February 2019). *mGlu5-KO* mice were raised on a C57BL/6 background (after backcrossing for more than ten generations) and were housed at the Florey Institute of Neuroscience and Mental Health. Adult male *mGlu5-KO* mice (aged 11–12 weeks) were anesthetized with 80 mg/kg pentobarbital prior to being killed via transcardial perfusion with 4% paraformaldehyde (PFA), in accordance with the Florey Institute Animal Ethics Committee (Ethics ID: 19-036, accepted in March 2019).

### 2.2. Calcium Imaging

Live Ca^2+^ imaging was performed on ex vivo preparations of the myenteric plexus from the proximal colon. After removal from the animal, the colon was pinned in a petri dish lined with silicone elastomer (Sylgard 184, Dow Corning, Midland, MI, USA), cut open along the mesenteric border, evenly stretched, and pinned flat with insect pins. Using fine forceps, the mucosa, submucous plexus, and longitudinal muscle were removed, leaving a circular muscle myenteric plexus (CMMP) preparation. A small inox ring was placed under the tissue, and a rubber O-ring was stretched over the top, securing the tissue across the ring [28]. This ring of CMMP was transferred to a glass-bottomed petri dish and constantly perfused with control Krebs solution at room temperature to reduce contractions. Imaging was performed using an inverted microscope (Axiovert 25, Zeiss, Jena, Germany) and × 20 (NA 0.5) objective lens with an Axiocam 702 mono camera (Zeiss, Jena, Germany). Videos 1 min in duration were acquired at 2 Hz (image size 512 × 512 pixels). Cells were illuminated with an LED (Zeiss Colibri) filtered at 470 nm with an exposure time of 30–50 msec/image.

Electrical stimulation was delivered by placing a focal stimulating electrode (Tungsten wire, 50 µm in diameter) over an internodal strand leading to a ganglion of interest. A single pulse (1P, 300 µs) followed by a train of 20 pulses (20P, 300 µs, 20 Hz) were delivered 5 min apart. Tissue was first stimulated with 1P and 20P (referred to as ‘initial’ stimulation) to establish baseline responses, then perfused with a mGlu receptor antagonist for 10 min followed by another sequence of 1P and 20P stimulation. Time control recordings were also obtained using the same protocol as the control Krebs solution during the “drug” phase. 1P stimulation was chosen to evoke fast synaptic transmission, while 20P stimulation evokes slow synaptic transmission. For the application of agonists, drugs were applied for 10 s onto a ganglion of interest while a 1 min video was recorded. All recordings were performed in Krebs solution + nicardipine (1.25 µM), and all agonists and antagonists were diluted in the same solution. At the end of the experiment, preparations were removed from the rings, pinned in a small, Sylgard-lined dish, and fixed overnight in 4% formaldehyde at 4 °C in preparation for post-hoc immunohistochemistry.

Ca^2+^ imaging videos were analyzed using Igor Pro software (Wavematrics, Lake Oswego, OR, USA), using custom-written routines [28]. A region of interest (ROI) was drawn over the cell body of each responding neuron to calculate the average fluorescent intensity, which was normalized to the baseline fluorescence for that cell (F/F_0_). The maximum increase in fluorescence was calculated from baseline to the peak amplitude (ΔF/F_0_) and compared between cells across different conditions. Recordings were made from a minimum of N = 3 mice for each agonist or antagonist. 

### 2.3. Video Imaging of Ex Vivo Colonic Motility

Colonic motility was examined in ex vivo preparations using a video imaging setup that has previously been described by our laboratory (Figure 1) [29]. To isolate the colon, an incision was made along the abdominal midline of the mouse using small spring scissors. The entire colon was removed and immediately placed in Krebs’ solution (composition in mM: 117 NaCl, 4.6 KCl, 2.5 CaCl2, 1.2 MgSO_4_, 1 NaH_2_PO_4_, 25 NaHCO_3_, 11 D-glucose) and bubbled with carbogen (95% O_2_, 5% CO_2_). The dissected colon was placed in an organ bath containing Krebs solution and heated to 35–37 °C. The colon was cannulated at the oral end, and any fecal content present in the colon was flushed out by increasing the pressure in the oral luminal inflow reservoir prior to cannulating the anal end. Both the oral and anal luminal inflow reservoirs were used to control the pressure within the colon. Excess mesentery remaining on the colon was carefully trimmed. A video camera (Logitech Quickcam Pro 9000, Logitech Carl Zeiss, Macquarie Park, NSW, Australia) connected to a computer with video acquisition software was positioned above the organ bath, capturing videos of the colon as it contracted at a rate of 30 frames per second.

Video imaging experiments began with a 45-min equilibration period, after which four videos (each 15 min in duration) were acquired under control conditions. Group-I mGlu receptor antagonists were then applied to the organ bath via a separate inflow reservoir, using a peristaltic pump to recirculate the antagonist solution (Figure 1). All antagonists were perfused through the organ bath for 10 min before four 15 min videos were captured. Time control experiments were also performed to confirm that the colon was unaffected by solution recirculation. Drugs were then washed out, reverting to an inflow of Krebs for 10 min, and four 15 min videos were captured.

The videos were obtained in a .avi format to allow conversion into a spatiotemporal heatmap (also known as diameter maps or D-maps), using in-house software ‘Scribble2′ that detects the outer edges of the colon and calculates the distance between these edges (i.e., the diameter of the gut) over time. Spatiotemporal heatmaps were analyzed using Analyse2 (2014) with Matlab software (R2017b) for (i) the number of colonic motor complexes (CMCs) per video, (ii) the length of CMCs, and (iii) the speed of CMCs. For each condition, the measurement of parameters was averaged from the four recordings. Contractions were considered CMCs if they propagated more than 50% of the length of the colon [29,30].

### 2.4. Pharmacological Agents

BAY 36-7620 (10 µM) was used as a negative allosteric modulator against mGlu1; MPEP (10 µM) was used as a negative allosteric modulator against mGlu5; two antagonists against Group-I mGlu receptors were also used, (RS)-3-aminoindan-1,5-dicarboxylic acid (AIDA, 500 µM) and N-phenyl 7-(hydroxyamino)cyclopropa[b]chromen-1a-carboxamide (PHCCC, 30 µM). (RS)-3,5-Dihydroxyphenylglycine (DHPG, 30 µM) was applied as an agonist of Group-I mGlu receptors. All drugs were purchased from Tocris Bioscience (via In Vitro Technologies, Noble Park North, VIC, Australia). BAY 36-7620, MPEP, and PHCCC were dissolved in DMSO, AIDA was dissolved in 1.1 eq NaOH, and DHPG was dissolved in distilled water.

### 2.5. RNAscope In Situ Hybridization

Tissue was collected from the mid colon of C57BL/6 mice, opened along the mesenteric border, pinned in a petri dish lined with silicon elastomer, and fixed overnight in 4% formaldehyde at 4 °C. The tissue was rinsed three times in PBS at 10 min intervals. The mucosa, submucous plexus, and circular muscle were carefully removed, leaving longitudinal muscle myenteric plexus (LMMP) preparations. The RNAscope^®^ 2.0 Assay (Advanced Cell Diagnostics, Newark, CA, USA, Wang et al., 2012) was used in conjunction with immunofluorescence staining [31]. This involved permeabilizing LMMP preparations in protease IV in a sealed, humidified container for 30 min at room temp and rinsing the preparations twice in PBS. RNAscope probes for mGlu1 (*Grm1*: Mm-Grm1, Cat#ADV449781), mGlu5 (*Grm5*: Mm-Grm5-C2, Cat#ADV423631C2), and positive and negative control probes (dapB) were warmed to 40 °C for 10 min (Appendix A). Tissue preparations were placed in 20 µL of probe solution for 2 h at 40 °C. After washing tissue preparations in wash buffer twice for 1 min, preparations were placed in 30 µL Amp 1-FL for 30 min at 40 °C, 30 µL Amp 2-FL for 15 min at 40 °C, 30 µL Amp 3-FL for 30 min at 40 °C, and 30 µL Amp 4-FL Alt-A/Alt-B for 15 min at 40 °C rinsing preparations in wash buffer between each step for 2× 1 min. After a final wash in wash buffer for 2× 1 min, preparations were washed in PBS 3× 1 min and incubated in 1% Triton for 30 min at room temperature. Tissue was then rinsed in PBS 3× 2 min and incubated in primary antisera and secondary antisera, as outlined below.

### 2.6. Immunofluorescence Staining

Tissue was fixed in 4% formaldehyde overnight at 4 °C, then rinsed for 3 × 10 min in PBS. For colon tissue from *mGlu5-KO* mice, the mucosa, submucosal plexus, and circular muscle were carefully peeled off, leaving the LMMP. All tissue was permeabilized in 10% CASBLOCK + 0.1% Triton for 30 min and incubated in primary antisera overnight at 4 °C. Tissue was then rinsed in PBS 3 × 10 min, incubated in secondary antisera for 2 h at room temperature, then rinsed again 3× 10 min in PBS prior to being mounted on glass slides in Dako fluorescence mounting medium (Dako, Carpentaria, CA, USA) and secured with a glass coverslip.

Primary antisera used included Human anti-HuC/D (Hu; 1:5000, gift from Dr V. Lennon), Sheep anti-nNOS (1:2000, gift from P. Emson), Rabbit anti-Calbindin (1:1600, Swant #CB-38a), Goat anti-Calretinin (1:1000, Swant #CG1), and Rabbit anti-S100β (1:1000, Dako #Z0311). Secondary antisera included donkey anti-Human Alexa Fluor^®^ 647 (1:500, Jackson #709-605-149), donkey anti-Sheep Alexa Fluor^®^ 488 (1:400, Molecular Probes #A11015) and Alexa Fluor^®^ 647 (1:500, Molecular Probes #A21448), and donkey anti-Rabbit Alexa Fluor^®^ 594 (1:400, Molecular Probes #A21207) and Alexa Fluor^®^ 647 (1:400, Molecular Probes #A31573).

### 2.7. Imaging and Analysis

Wholemount preparations processed for RNAscope and immunofluorescence staining were imaged using an LSM880 confocal microscope (Carl Zeiss Microscopy, North Ryde, NSW, Australia). Multi-channel Z-stack images were captured with a Plan-Apochromat 63x/1.4 Oil DIC M27 objective lens, and samples were excited with Diode lasers at wavelengths 488, 561, and 633. Images were 1024 × 1024 pixels in size with a file depth of 8 bits, and Z-stacks were captured at 0.5 µm intervals. Confocal images were converted to .ims files for analysis with Imaris 9.0 Image Analysis Software (Bitplane, Belfast, UK), as previously outlined [31]. Briefly, this involved creating 3D reconstructions of neuronal and glial surfaces, as well as individual mRNA puncta, and isolating the mRNA that was directly contained within the neurons or glia. The number of mRNA puncta within each cell was counted and analyzed.

Wholemount preparations from *mGlu5-KO* mice were imaged with a Zeiss Axio Imager2 fluorescence microscope (Carl Zeiss Microscopy, North Ryde, NSW, Australia). 4–5 randomly selected areas per region (i.e., proximal, mid, and distal colon) were imaged with a 20× objective lens. For cell count analysis, multi-channel images were analyzed in Image J/FIJI using the ‘Cell Counter’ plugin, as previously outlined [32]. For each image, all Hu+ neurons were counted, and the numbers of nNOS+ and CalB+ neurons were expressed as a proportion of the Hu+ neurons.

### 2.8. Data Analysis

All data are shown as mean ± SEM. Statistical analysis was performed using Graphpad Prism (Version 5.04, San Diego, CA, USA). Video imaging data were analyzed using a paired *t*-test. RNAscope, calcium imaging, and cell count data were analyzed using the Students’ *t*-test (two-tailed, unpaired *t*-test) and other analyses using one-way ANOVA.

## 3. Results

### 3.1. Group-I mGlu Receptors Are Widely Expressed in the Myenteric Plexus

To examine whether mGlu1 and mGlu5 transcripts were expressed by specific cellular subtypes in the myenteric plexus, RNAscope in situ hybridization was performed to identify the location of individual *Grm1* and *Grm5* mRNA, followed by immunofluorescent staining. Tissues from at least three mice per group were investigated. Almost all neurons and glial cells examined contained copies of either *Grm1* or *Grm5* mRNA. HuC/D (Hu) is a pan-neuronal marker in the ENS. We identified that 98% of Hu+ neurons analyzed contained *Grm1* mRNA (*n* = 197/202 cells), and 99% of Hu+ neurons expressed *Grm5* (*n* = 200/202 cells). Nitric oxide is a major inhibitory neurotransmitter, and neuronal nitric oxide synthase (nNOS) is expressed by inhibitory motor neurons, as well as a small population of interneurons. Of 104 nNOS+ neurons analyzed, all contained *Grm1*, and 103 contained *Grm5* (99%) mRNA. 87 out of 96 S100β+ enteric glia expressed *Grm1* (91%) and 91/96 S100β+ glia expressed *Grm5* (95%). In Hu+ neurons, nNOS+ neurons, and S100β+ glia, we identified a greater number of copies of *Grm5* mRNA compared to *Grm1* (Figure 2).

### 3.2. Group-I mGlu Receptors and Myenteric Neuron Activity

Live Ca^2+^ imaging was performed to investigate the influence of mGlu receptors on neuronal activity. The Group-I mGlu receptor agonist DHPG produced intracellular calcium ([Ca^2+^]_i_) transient in *n =* 80/154 neurons (from *N* = 3 mice; Figure 3A). Post hoc immunofluorescence was performed to examine whether responsive cells were immunoreactive for calbindin (CalB), which is mostly used as a marker of intrinsic sensory neurons of the ENS and nNOS. Both calbindin+ and nNOS+ neurons responded to DHPG. Interestingly, we observed some myenteric neurons that were immunoreactive against both calbindin+ and nNOS+ (CalB+/nNOS+), which have not been previously described in the mouse ENS [33], which were also responsive to DHPG. There was no significant difference in the response amplitude between the different neuronal phenotypes (Figure 3C,D). Some GCaMP6f+ non-neuronal cells also responded to the DHPG application. Although specific immunohistochemistry was not performed, the morphology of these cells resembled that of enteric glia (Figure 3B,E).

To examine the involvement of Group-I mGlu receptors in native myenteric synaptic transmission, we investigated the effects of various antagonists on responses to electrical stimulation of interganglionic fiber tracts. Both single pulse (1P) stimulation, which evokes fast transmission, and train (20P) stimulation, which also evokes slow transmission, were applied. Changes in the amplitude of [Ca^2+^]_i_ transients were examined in the presence of various Group-I mGlu receptor inhibitors. The non-specific Group-I mGlu receptor antagonist AIDA reduced [Ca^2+^]_i_ transients to both single and train pulse stimulation, suggesting that these receptors are involved in both fast and slow excitatory neurotransmission (Figure 4). However, while the application of the mGlu5 specific antagonist MPEP increased [Ca^2+^]_i_ transient amplitude, the mGlu1 specific antagonist BAY36-7620 had no significant effect on neuronal responses to either 1P or 20P stimulation (Figure 4). Application of a different Group-I mGlu receptor antagonist, PHCCC, led to increased neuronal responses following 1P stimulation (Figure 4D) but not responses to 20P stimulation (Figure 4H).

**Figure 2 biomolecules-13-00139-f002:**
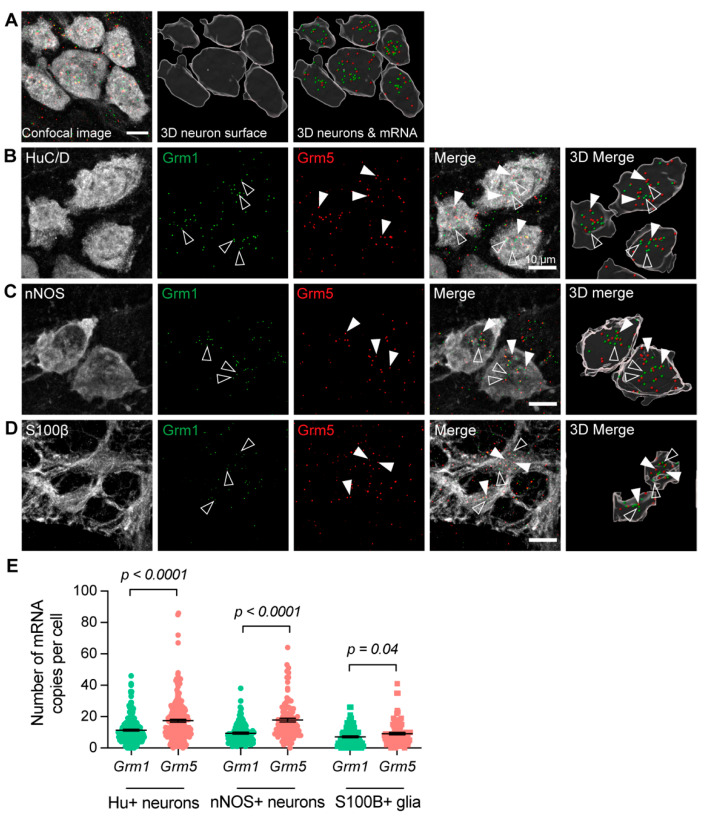
**Expression of *Grm1* and *Grm5* in the myenteric plexus of the mouse mid colon.** (**A**) Representative images showing an original confocal micrograph of HuC/D+ (Hu+) neurons together with *Grm1* (green) and *Grm5* (red) mRNA signals (left), 3D-rendered surfaces of the Hu+ neurons (middle), and 3D-rendered surfaces of *Grm1* and *Grm5* mRNA signals together with Hu+ neurons (right). Each spot depicts a single mRNA transcript of *Grm1* (arrow) and *Grm5* (open arrow). (**B**–**D**) Panels showing *Grm1* and *Grm5* expression in Hu+ neurons (**B**), nNOS+ neurons (**C**), and S100B+ glia (**D**). In each panel, the confocal micrographs of the immunofluorescent signals are shown on the left, with the 3D-rendered cells on the right. *Grm1* transcripts are indicated by open triangles, and *Grm5* transcripts by closed triangles. (**E**) Average number of individual mRNA for *Grm1* and *Grm5* in Hu+ neurons, nNOS+ neurons, and S100β+ glia. 197/202 (98%) Hu+ neurons expressed *Grm1* (11.4 ± 0.6 *Grm1* mRNA copies per Hu+ neuron, *N* = 3 mice), and 200/202 (99%) of Hu+ neurons expressed *Grm5* (17.4 ± 0.9 *Grm5* mRNA copies per Hu+ neuron, *N* = 3 mice). All 104 nNOS+ neurons analyzed contained Grm1 (9.4 ± 0.6 *Grm1* mRNA copies per nNOS+ neuron), and 103/104 (99%) contained Grm5 (17.9 ± 1.2 *Grm5* mRNA copies per nNOS+ neuron, *N* = 3 mice). 87/96 (91%) S100β+ glia expressed *Grm1* (7.2 ± 0.6 *Grm1* mRNA copies per S100β+ glia, *N* = 3) and 91/96 (95%) S100β+ glia expressed *Grm5* (9.2 ± 0.8 *Grm5* mRNA copies per S100β+ glia, *N* = 3). Student’s *t*-test.

**Figure 3 biomolecules-13-00139-f003:**
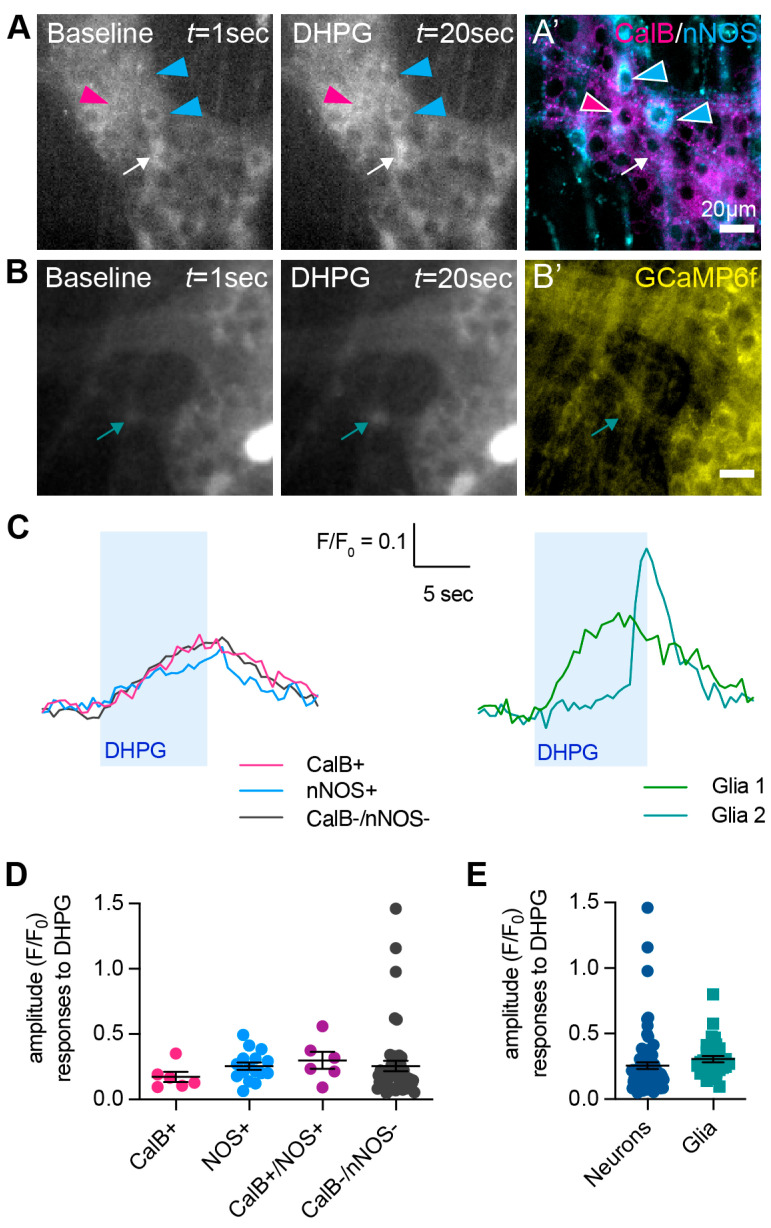
**Responses to the application of the Group-I mGlu receptor agonist (RS)-3,5-DHPG.** (**A**,**B**) Representative images of GCaMP6f fluorescence (grey) during live Ca^2+^ recording of a myenteric ganglion at baseline (left, *t* = 1 s) and after application of DHPG (right, *t* = 20 s). (**A′**) *Post-hoc* immunohistochemistry was performed on the same ganglion against calbindin (CalB+, magenta) and nNOS+ (cyan). Neurons responding to DHPG were subsequently identified to be CalB+ (magenta triangle), nNOS+ (cyan triangles), CalbB+/nNOS+ (not shown), or neither CalB+ nor nNOS+ (CalB−/−nNOS−; white arrow). (**B′**) Post-hoc immunohistochemistry against GCaMP6f (yellow). Some DHPG-responsive cells had the morphology of enteric glia and, in some cases, were extra-ganglionic (green arrow). (**C**) Representative traces from a CalB+ neuron (magenta), nNOS+ neuron (blue), and CalB-/nNOS-, responding to the application of DHPG (left). Representative traces from two presumptive glial cells responding to the application of DHPG (right). (**D**) Quantification of [Ca^2+^]_i_ transient amplitudes to DHPG in CalB+ neurons (ΔF/F_0_ 0.17 ± 0.04, *n* = 6), nNOS+ neurons (ΔF/F_0_ 0.25 ± 0.03, *n* = 16), CalB+/nNOS+ neurons (ΔF/F_0_ 0.30 ± 0.07, *n* = 6), and CalB−/nNOS− neurons (ΔF/F_0_ 0.26 ± 0.04, *n* = 49). (**E**) Quantification of [Ca^2+^]_i_ amplitudes to DHPG in neurons overall (ΔF/F_0_ 0.26 ± 0.03, *n* = 80) vs. presumptive glial cells (ΔF/F_0_ 0.31 ± 0.02, *n* = 34). Scalebars = 20 µm.

**Figure 4 biomolecules-13-00139-f004:**
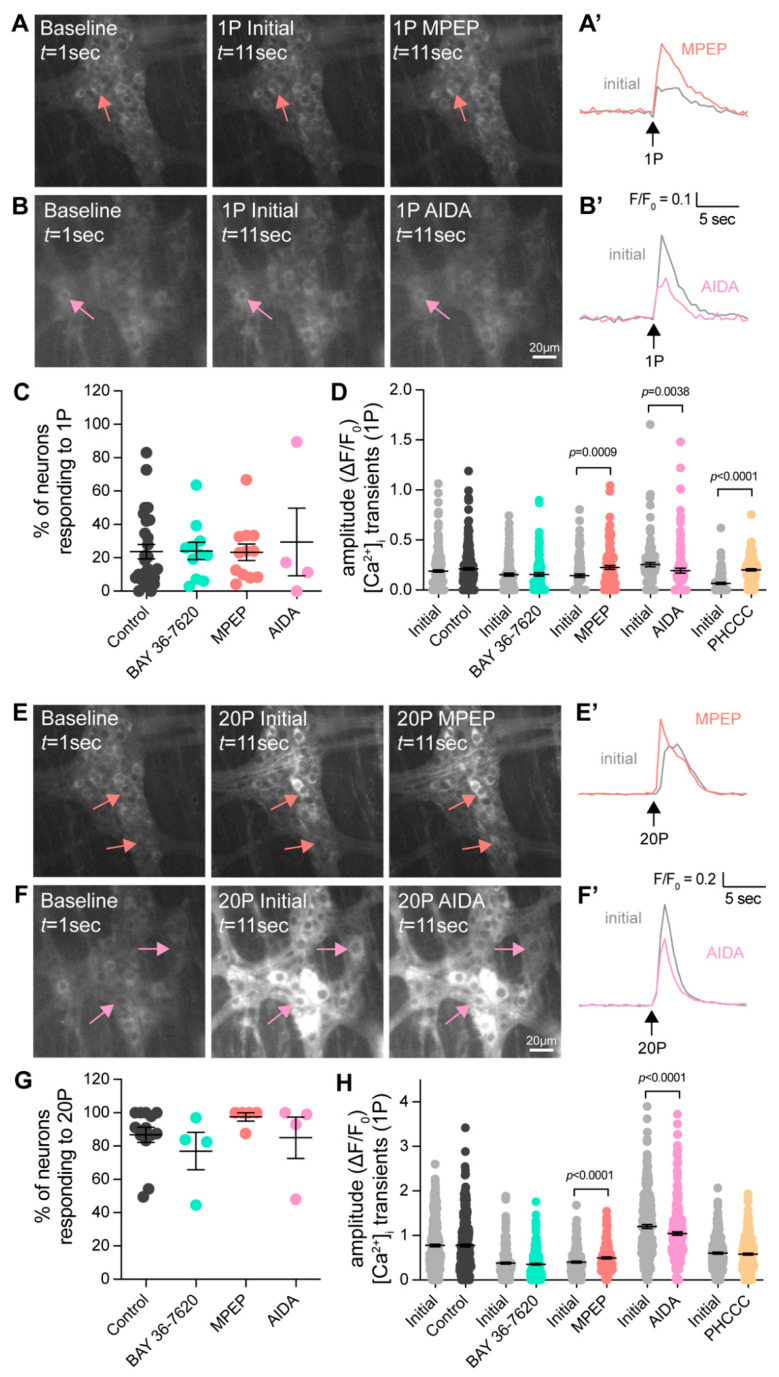
**Effect of Group-I mGlu receptor antagonists on enteric neurotransmission following electrical stimulation.** (**A**,**B**) Representative images of GCaMP6f fluorescence during live Ca^2+^ recording of a myenteric ganglion at baseline (left, *t* = 1 s) after single (1P) stimulation of an interganglionic fiber tract in the presence of control Krebs’ solution (Initial, middle, *t* = 11 s) and after application of an antagonist for 10 min (right, *t* = 11 s). Arrows indicate neurons responding to electrical stimulation (different colours correspond to experiments in different antagonists). (**A′**) Representative traces from a single neuron during Initial electrical stimulation and after MPEP application, resulting in an increase in [Ca^2+^]_i_ transient amplitude. (**B′**) Representative traces from a single neuron during Initial electrical stimulation and after AIDA application, resulting in a decrease in [Ca^2+^]_i_ transient amplitude. (**C**) No difference in the proportion of neurons responding to 1P stimulation (Time Control: 23.7 ± 4.3%; BAY 36-7620: 24.1 ± 5.2%; MPEP: 23.3 ± 5.0%; AIDA: 29.5 ± 20.3%; one-way ANOVA). (**D**) Quantification of [Ca^2+^]_i_ transient amplitude following 1P stimulation in the presence of various Group-1 mGlu receptor antagonists. [Ca^2+^]_i_ responses were significantly increased in the presence of MPEP (10 µM. Initial: ΔF/F_0_ 0.14 ± 0.02, *n* = 107; MPEP: ΔF/F_0_ 0.23 ± 0.02, *n* = 108, *p* = 0.0009) and PHCCC (30 µM. Initial: ΔF/F_0_ 0.07 ± 0.01, *n* = 117; PHCCC: ΔF/F_0_ 0.20 ± 0.01, *n* = 117, *p* < 0.0001). AIDA significantly decreased responses (500 µM. Initial: ΔF/F_0_ 0.21 ± 0.02, *n* = 77; AIDA: ΔF/F_0_ 0.01 ± 0.02, *n* = 78, *p* < 0.0001). No significant changes were observed in Time Controls or BAY 36-7620 (*p* > 0.05; Students *t*-test for all). (**E**,**F**) Representative images during train (20P) stimulation in the presence of control Krebs’ solution (Initial, middle, *t* = 11 s) and after application of an antagonist for 10 min (right, *t* = 11 s). Traces from a single neuron during Initial electrical stimulation and after MPEP application (**E′**), resulting in an increase in [Ca^2+^]_i_ transient amplitude, and after AIDA application (**F′**), resulting in a decrease in [Ca^2+^]_i_ transient amplitude. (**G**) No difference in % of neurons responding to 20P stimulation (Con: 86.8 ± 4.6%; BAY 36-7620: 77.0 ± 11.3%; MPEP: 97.5 ± 2.5%; AIDA: 85.0 ± 12.4%; one-way ANOVA). (**H**) [Ca^2+^]_i_ transient amplitudes to 20P stimulation were increased following the application of MPEP (Initial: ΔF/F_0_ 0.40 ± 0.02, *n* = 158; MPEP: ΔF/F_0_ 0.50 ± 0.02, *n* = 158, *p* < 0.0001), decreased after the application of AIDA (Initial: ΔF/F_0_ 1.10 ± 0.04, *n* = 231; AIDA: ΔF/F_0_ 0.85 ± 0.03, *n* = 231, *p* < 0.0001). No significant changes in response to 20P stimulation were seen in Time controls, after application of BAY 36-7620 or PHCCC (*p* > 0.05, Students *t*-test for all).

Post-hoc immunohistochemistry was also performed to examine whether inhibition of mGlu receptors differed in various subtypes of enteric neurons. For the most part, neurons that were immunoreactive for either calbindin, nNOS, or both calbindin and nNOS (CalB+/nNOS+) did not exhibit differences in their electrically evoked [Ca^2+^]_i_ amplitudes in the presence of BAY 36-7620, MPEP, or AIDA compared to initial responses (Figure 5). Calbindin+ neurons did exhibit increased responses to MPEP, as well as neurons that were immunoreactive for neither calbindin nor nNOS (CalB−/nNOS−; Figure 5D). Interestingly, CalB−/nNOS− neurons also exhibited reduced responses in the presence of the Group-I mGlu receptor antagonist, AIDA (Figure 5E). Overall, this suggests that mGlu5 transmission occurs on intrinsic sensory neurons, but Group-I mGlu receptor transmission does not occur on inhibitory motor neurons. The identity of the CalB−/nNOS− neurons remains to be investigated further but could include excitatory motor neurons as well as populations of interneurons.

### 3.3. Group-I mGlu Receptors Are Involved in Initiation of Colonic Motility

To examine the role of Group-I mGlu receptors in colonic motility, the influence of the mGlu1 antagonist BAY 36-7620, the mGlu5 antagonist MPEP, as well as the combined Group-I mGlu receptor antagonists AIDA and PHCCC on colonic motor complexes (CMCs) were investigated. Significant decreases in CMC frequency were observed following the application of BAY 36-7620 (Figure 6B) and PHCCC (Figure 6E). There were no changes to CMC length or speed. No changes to any CMC parameters were observed in Time controls (Figure 6A) or following the application of MPEP (Figure 6C) or AIDA (Figure 6D).

**Figure 5 biomolecules-13-00139-f005:**
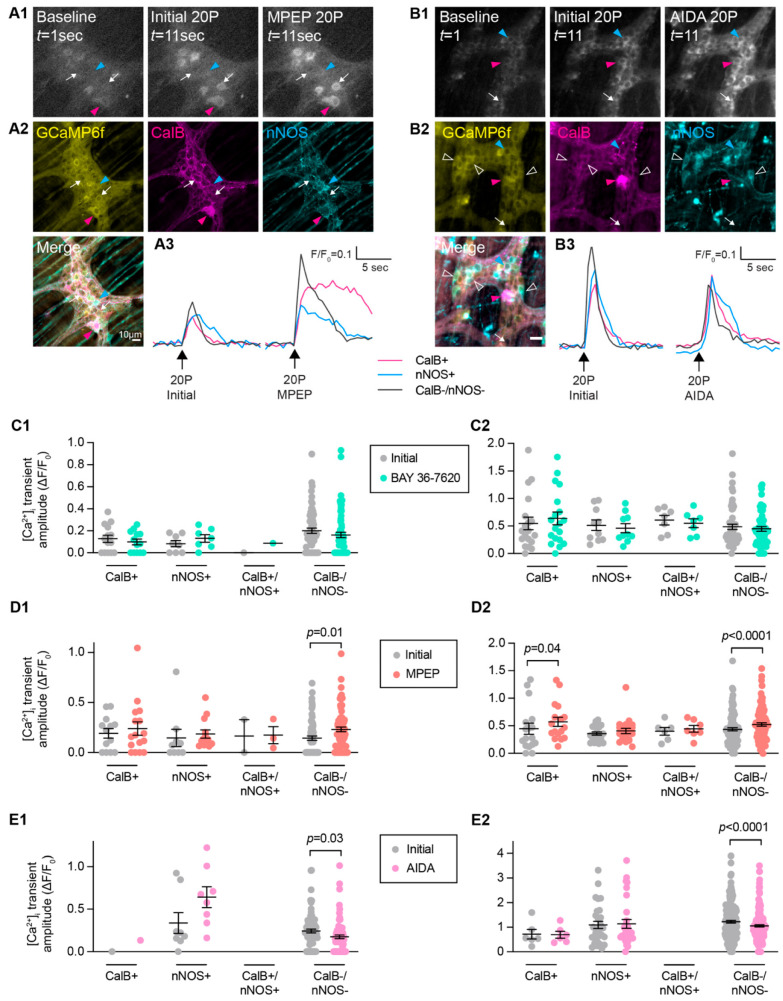
**Effect of Group-I mGlu receptor antagonists in different neuronal subtypes.** (**A1,B1**) Representative images of GCaMP6f fluorescence during live Ca^2+^ recording of a myenteric ganglion at baseline (left, *t* = 1 s) after single (20P) stimulation of an interganglionic fiber tract in the presence of control Krebs’ solution (Initial, middle, *t* = 11 s) and after antagonist application (right, *t* = 11 s). (**A2**,**B2**) *Post-hoc* immunofluorescence staining of the same ganglion for GCaMP6f (yellow), Calbindin (CalB, magenta), and nNOS (cyan). Responsive neurons were found to be CalB+ (magenta triangle), nNOS+ (cyan triangle), or CalB−/nNOS− (white arrow) (**A3**,**B3**). Representative traces from a CalB+ neuron (magenta), nNOS+ neuron (blue), and CalB−/nNOS− neuron (dark grey) responding to 20P stimulation initially and following antagonist application. (**C1**,**C2**) No difference in [Ca^2+^]_i_ transient amplitude (ΔF/F_0_) following 1P or 20P stimulation between Initial stimulation and in the presence of the mGlu1 antagonist BAY 36-7620 (10 µM) was observed in CalB+, nNOS+, CalB+/nNOS+, or CalB−/nNOS− neurons. (**D1**,**D2**) MPEP triggered significant increases in [Ca^2+^]_i_ transient amplitude in CalB-/nNOS- neurons following both 1P (Initial: ΔF/F_0_ 0.14 ± 0.02, *n* = 71; MPEP: ΔF/F_0_ 0.23 ± 0.02, *n* = 71) and 20P stimulation (Initial: ΔF/F_0_ 0.44 ± 0.03, *n* = 98, 20P MPEP: ΔF/F_0_ 0.53 ± 0.03, *n* = 98). (**E1**,**E2**) AIDA triggered decreased responses in CalB−/nNOS− neurons following 1P (Initial: ΔF/F_0_ 0.24 ± 0.02, *n* = 80; AIDA: ΔF/F_0_ 0.17 ± 0.02, *n* = 80) and 20P stimulation (Initial: ΔF/F_0_ 1.23 ± 0.05, *n* = 179; AIDA: ΔF/F_0_ 1.06 ± 0.05, *n* = 179). Student’s *t*-test for all. Scale bars = 10 µm.

### 3.4. mGlu5-KO Mice Show No Change in Myenteric Neurochemistry

As mGlu receptor activity is involved in the colonic function and mGlu5 expression was observed in almost all enteric neurons, we investigated whether there are changes to the ENS in *mGlu5-KO* mice compared to wild-type mice. Immunohistochemistry was performed on the proximal, mid, and distal colon against Hu, calbindin, and nNOS. Overall, we observed no significant changes in the number of Hu+ neurons, or the proportions of either calbindin+, nNOS+, or Calb+/nNOS+ neurons in *mGlu5-KO* mice compared to controls (Figure 7).

## 4. Discussion

In this study, we showed that Group-I mGlu receptors are involved in myenteric neurotransmission and that they may be involved in colonic motility. Using RNAscope in situ hybridization, we confirmed that *Grm1* and *Grm5* are expressed in the ENS of mice and found that both transcripts were present in nearly all myenteric neurons and glia. This extends previous studies showing that Group-I mGlu receptors are expressed widely in the ENS of other experimental species [16,23]. On average, individual cells contained greater numbers of *Grm5* mRNA transcripts than *Grm1* mRNA transcripts (as estimated by RNAscope), which could suggest that mGlu5 plays a larger role in ENS function than mGlu1. This is supported by our calcium imaging data, showing that the mGlu5 antagonist MPEP had a significant effect on electrically evoked [Ca^2+^]_i_ transient amplitudes, while the mGlu1 antagonist BAY 36-7620 did not. Interestingly, while *Grm1* and *Grm5* mRNAs were identified in almost all myenteric neurons and glia, only half (*n* = 80/185) of Hu+ enteric neurons responded to the application of the Group-I mGlu receptor agonist (RS)-3,5-DHPG. This highlights that while the genes encoding mGlu receptors may be expressed by all cells, the amount of functional protein produced remains to be investigated further. In addition, alternative splicing of *Grm1* and *Grm5* is known to take place in CNS neurons, resulting in different variants of mGlu1 and mGlu5 that may differ in their function [34]. Which splice variants predominate in the ENS have not been investigated.

Responsive neurons included intrinsic sensory neurons (calbindin+), inhibitory motor neurons (nNOS+), as well as cells immunoreactive for both nNOS+/CalB+ and neither nNOS−/CalB−, confirming that Group-I mGlu receptors are active in a number of different subtypes of enteric neurons. However, some of these responses may have been produced by indirect activation via synaptic communication. An additional issue is that while [Ca^2+^]_i_ transients were evoked by (RS)-3,5-DHPG, how this relates to membrane potential change and action potential firing remains to be investigated. Our recent studies show that [Ca^2+^]_i_ transients are evoked in both AH- and S-electrophysiological types of enteric neurons following stimulation by short-duration depolarization and also by sub-threshold changes in membrane potential [35].

### 4.1. The Role of Group-I mGlu Receptors on Synaptic Transmission and Colonic Motility

Previously, electrophysiological recordings in guinea pig ENS have shown that Group-I mGlu receptors contribute to slow excitatory postsynaptic potentials in both myenteric and submucous plexus neurons [23,24,36]. Our live calcium imaging data show that inhibition of mGlu5 using MPEP produced increased [Ca^2+^]_i_ responses to electrical stimulation. As the activation of Group-I mGlu receptors has generally been reported to trigger excitatory postsynaptic transmission and increased [Ca^2+^]_i_ transients usually suggest increased excitability; it was therefore unexpected that inhibition of mGlu5 led to an increase in neuronal excitability. In the CNS, Group-I mGlu receptors interact with many different channels and synaptic proteins resulting in multiple different effects on synaptic transmission and excitability of a cell [37,38,39]. Although most studies show that Group-I mGlu receptors facilitate neuronal depolarization and potentiation of excitatory responses, they can also produce inhibitory effects. For example, in midbrain dopaminergic neurons, Group-I mGlu receptor activation results in an inhibitory postsynaptic response via activation of a Ca^2+^-dependent K^+^ current [40]. This same apamin-sensitive K^+^ current has been recorded from enteric neurons [41]; however, whether it can be modulated by Group-I mGlu receptors in the ENS remains to be examined. It is also possible that mGlu5 is present in inhibitory enteric neurons, and thus inhibition of mGlu5 transmission results in increased neuronal excitability in the network. Post-hoc immunohistochemistry showed that MPEP application altered transmission to calbindin+ as well as calbindin-/nNOS− cells, suggesting that mGlu5 is acting on intrinsic sensory neurons as well as populations of interneurons and possibly excitatory motor neurons. Further investigation into the neurochemistry of these groups is needed; however, due to the scarcity of specific labels for individual subtypes of enteric neurons, this is currently difficult to accomplish. In addition, the application of MPEP did not alter the frequency of colonic migrating complexes, suggesting that while mGlu5 transmission is present in enteric neuron-neuron communication, it is not involved in the generation of CMCs.

On the other hand, inhibition of mGlu1 transmission using BAY 36-7620 did not significantly affect electrically-evoked [Ca^2+^]_i_ transients but did reduce the frequency of CMCs. The additional complicating factor in colonic motility investigations is that receptors outside the ENS could also be implicated. Although specific cell types have not been conclusively identified in situ in the mouse, Group-I mGlu receptor equivalents are expressed by intestinal epithelial stem cells in *Drosophila melanogaster* [42], and mGlu1 expression has been shown in an enteroendocrine cell line as well as homogenates of the mouse intestinal mucosa [43]. Therefore, it is possible that BAY 36-7620 influences CMC frequency by inhibiting mGlu1 activity on non-neuronal cells.

To add another layer of complexity, the application of the combined Group-I mGlu receptor antagonist AIDA decreased electrically-evoked [Ca^2+^]_i_ transients but did not affect CMC frequency. One possibility is that AIDA is simultaneously blocking two different circuits: non-neuronal mGlu1 transmission and inhibitory circuits involving mGlu5 transmission, which cancel each other out. In addition, studies have highlighted that mGlu1 and mGlu5 can have different roles within the same neurons [44] and, therefore, could have different involvement in CMC generation. A different non-specific Group-I mGlu receptor antagonist PHCCC produced increased [Ca^2+^]_i_ transients following 1P electrical stimulation, did not affect train (20P) stimulation, and decreased CMC frequency. Both drugs have previously been used and shown to be effective in ENS. PHCCC reduces slow EPSP amplitude following stimulation of fiber tracts using both intracellular electrophysiology and calcium imaging recordings [25,36]. AIDA has also been used to reduce the amplitude of depolarization responses to glutamate in the ENS [24]. However, while both can function as a combined Group-1 mGlu receptor antagonist, whether there are fewer specific actions by each drug is unknown. PHCCC has also been identified to be a positive allosteric modulator for mGlu4 [45]. mGlu4 expression has not been identified on enteric neurons but is present in intestinal tissue from the stomach and duodenum [46] and could be present in the colon [47].

Constitutive activity of glutamatergic receptors could be a further compounding factor. There is evidence in the neonatal rodent brain that Group-I mGlu receptors are tonically active; however, this was absent from the adult brain [48]. Whether there is constitutive glutamatergic receptor activity in the ENS remains to be investigated further. It has been proposed that some serotonergic receptors (e.g., 5-HT3 receptors) in the gut behave in this fashion [49].

### 4.2. Group-I mGlu Receptors in Enteric Glia

Some of the changes above could also be attributed to the activity of enteric glial cells. Non-neuronal ENS cells in our *Wnt1-cre;GCaMP6f* mice also responded to DHPG with [Ca^2+^]_i_ transients. While we lack definitive post-hoc immunohistochemistry data, these cells had the morphological appearance of enteric glia. Enteric glial cells have previously been shown to respond to excitatory neurotransmitters with increases in [Ca^2+^]_i_ transients [50,51]. Expression of mGlu1/5 has been demonstrated in enteric glia previously [16], and this is also confirmed by our RNAscope data. In the CNS, Group-I mGlu receptors, particularly mGlu5, have been implicated in the “tripartite synapse”, which refers to the interaction between a presynaptic bouton, postsynaptic membrane, and an astrocytic process [52,53]. The activity of mGlu5 in astrocytes has been shown to be important for the modulation of postsynaptic activity. Specifically, activation of Group-I mGlu receptors leads to increased cytosolic Ca^2+^ in astrocytes [54,55]. The question of how inhibition of mGlu1/5 on enteric glia may influence neuronal communication is an avenue for further exploration. It has been established that enteric glia are also involved in colonic motility [56,57,58], but their role in modulating synaptic communication between enteric neurons has not been thoroughly explored. It is possible that mGlu receptor activity in enteric glia is the missing link that would explain the role of Group-I mGlu receptors in colonic activity. This area of study requires more detailed exploration in the future.

## 5. Conclusions

Our findings indicate that Group-I mGlu receptors may be involved in myenteric neuronal activity and colonic motility in mice. As mGlu1 and mGlu5 mRNA transcripts are present in myenteric glia, determining the role of their encoded glutamate receptor proteins in neuronal-glial and glial-glial communication will be useful in understanding their function, and broader functions of glutamatergic signaling, in the ENS. 

## Figures and Tables

**Figure 1 biomolecules-13-00139-f001:**
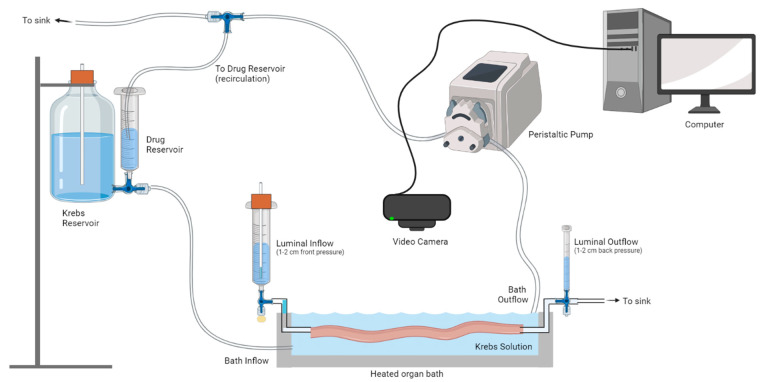
**Video imaging analysis of ex vivo colonic motility.** An isolated colon was placed in a heated organ bath and cannulated at the oral and anal ends. Luminal inflow was maintained at constant pressure, and fresh Krebs solution perfused the organ bath. A peristaltic pump was used to maintain a constant amount of Krebs in the organ bath. To conserve the number of antagonists used, they were recirculated from the reservoir via a peristaltic pump. Video recordings of colonic contractions were captured using a video camera connected to a computer. Four lots of 15 min recordings were made at each time point: (i) initial Krebs, (ii) antagonist (or time control Krebs), and (iii) washout. Figure made using Biorender.

**Figure 6 biomolecules-13-00139-f006:**
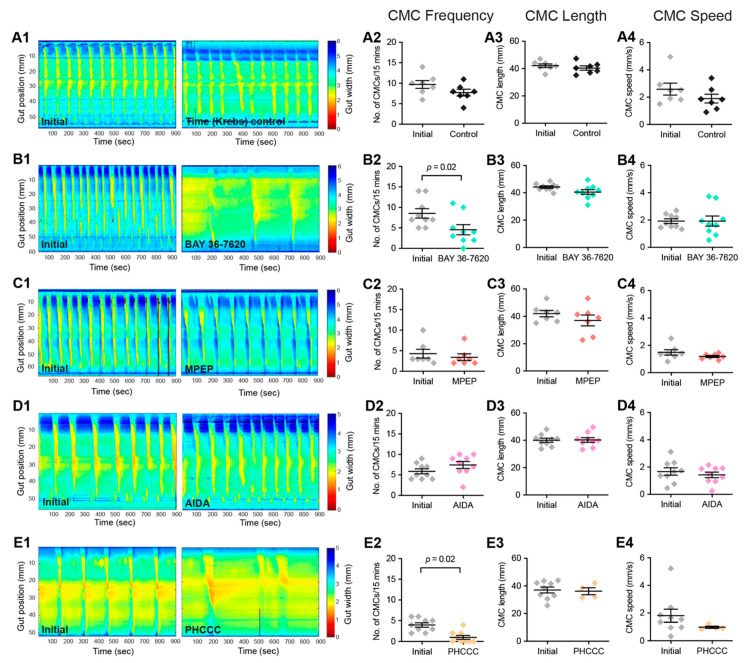
**Antagonists of Group-I mGlu receptors reduce colonic motility.** Spatiotemporal heat maps were generated from video recordings of colonic contractions ex vivo, showing changes in colonic diameter over time. The contractions in these maps were compared between Initial baseline recordings and the different drug conditions: (**A**) Time control, (**B**) BAY 36-7620 (10 µM), (**C**) MPEP (10 µM), (**D**) AIDA (500 µM), and (**E**) PHCCC (30 µM). (**A2**–**A4**) No significant difference in CMC frequency, length, or speed was observed during time control experiments. (**B2**–**B4**) In the presence of BAY 36-7620, a significant decrease in CMC frequency was observed (Control: 8 ± 1 CMCs/15 min; BAY 36-7620: 5 ± 1 CMCs/15 min, *n* = 9, *p* = 0.01); however, no difference was observed in (**A3**) CMC length or (**A4**) CMC speed. (**C2**–**C4**) MPEP had no effect on CMC frequency, length, or speed. (**D2**–**D4**) AIDA had no effect on CMC frequency, length, or speed. (**E2**) PHCCC led to a significant decrease in CMC frequency (Control: 4 ± 1 CMMCs/15 min; PHCCC: 1 ± 1 CMCs/15 min, *n* = 9, *p* = 0.002) but had no effect on (**E3**) CMC length, or (**E4**) CMC speed. Paired *t*-test for all.

**Figure 7 biomolecules-13-00139-f007:**
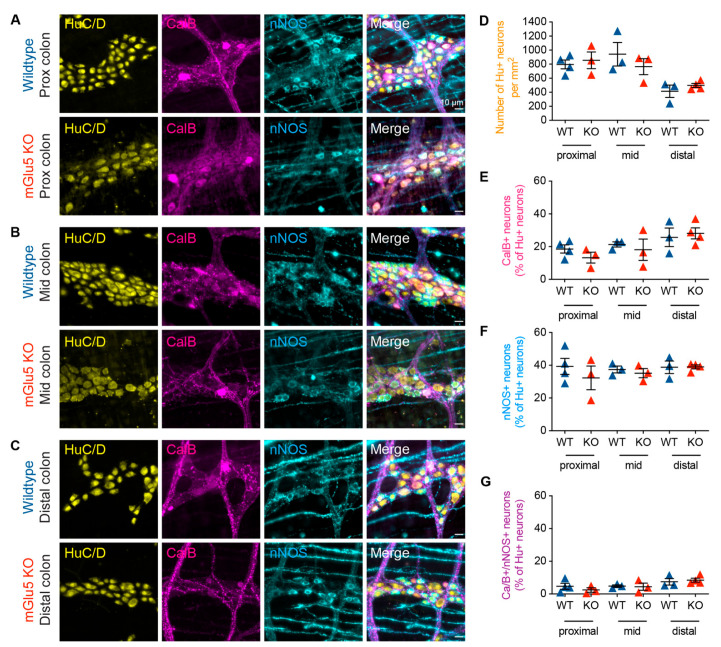
***mGlu5 KO* mice showed no alteration in proportions of myenteric neurochemical subtypes in the mouse colon.** (**A**–**C**) Representative images of immunofluorescence staining of the myenteric plexus of the mouse colon, showing labeling of the pan-neuronal marker Hu (yellow), CalB (magenta), and nNOS (yellow) in wildtype and *mGlu5 knock-out (KO)* mice in the proximal, mid, and distal colon. (**D**–**G**) Cell count analysis revealed no significant difference in the density of Hu+ neurons (per mm^2^), the proportion of CalB+ neurons (out of Hu+ neurons), the proportion of nNOS+, or the proportion of CalB+/nNOS+ double immunoreactive neurons. Scale bars = 10 µm. *p* < 0.05, *n* = 3–4 mice, Students’ *t*-test for all.

## Data Availability

All data from this study can be made available upon request. Please contact the corresponding authors.

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
