# Peer review of "Group I Metabotropic Glutamate Receptors Modulate Motility and Enteric Neural Activity in the Mouse Colon"

_biomolecules, 2023, doi:10.3390/biom13010139_

Round 1
Reviewer 1 Report
This study investigates the possible role of group I metabotropic receptors in the enteric nervous system and potentially in colonic motility in isolated mouse colon. The results are clear and convincing. The data is novel and useful to the field. The appropriate time controls for bleaching calcium signals appears to have been performed. The immunohistochemistry seems sound and spatio-temporal mapping data of CMCs.
My major concern is the speculation that mGlu receptors are involved in normal ongoing colonic motility (CMCs), because the antagonists of mGlu receptors inhibited CMCs. This statement is not adequately supported by the data and needs revision. An alternative possibility is raised below and needs discussion, with some extra citations, for balance.
For example, line 445 states “In this study, we showed that Group I-mGlu receptors are involved in myenteric neurotransmission and for the first time, their involvement in colonic motility.” This is a strong overstatement and needs to be toned down.
There is no doubt the authors data clearly shows that mGlu receptor inhibition with AIDA reduces the calcium transients evoked by nerve stimulation. Again, the authors interpret this as evidence that glutamate is involved in neurotransmission. Another possibility is that glutamate is not involved in neurotransmission, but the antagonists reduce constitutively active glutamateric receptor activity in the soma of enteric nerve cell bodies and this drop in membrane resistance reduces neuronal excitability for Ach to generate fast cholinergic EPSPs.
As the authors appreciate, visceral organs like the gut express a variety of receptors. Some may have a clear functional role, but many have no known function (at present). The fact that mGlu receptors are present in the ENS has been known for some time. But, finding a synaptic input using intracellular electrophysiology that is mediated by glutamate has not been easy.
No mention is made (that I could see) of the possibility of constitutively active glutamatergic receptors. This definitely needs inclusion in the discussion.
It is noteworthy to remember that numerous 5-HT receptors exist in the ENS, but finding a functional role for endogenous 5-HT in synaptic transmission in mouse ENS has not been easy, even though antagonists for 5-HT3 receptors potently block CMCs (Spencer et al. 2013) https://pubmed.ncbi.nlm.nih.gov/23593931/ This should be mentioned and cited.
Line 550. “Our findings indicate that…. are involved in myenteric neuronal activity” is again too strong. Please reword “Our findings indicate that….. maybe involved in myenteric neuronal activity…”
For example, in Figure 6 legend, it is stated “Group-I mGlu receptors are involved in colonic motility.” The data actually doesn’t fully support this. It certainly shows that the antagonists inhibit colonic motility. This is very different from implying that these receptors are involved in “normal” CMC activity.
The mGluR7 has been shown to exhibit constitutive activity. See: https://pubmed.ncbi.nlm.nih.gov/25881041/ This publication states “These findings introduce a novel potential physiological role for mGluR7 in the nervous system, that of a constitutively active receptor, and thereby suggest a model in which mGluR7 signaling may be impactful without the need to invoke strong receptor activation by millimolar concentrations of extracellular glutamate.” This reference should be quoted and mention made of the possibility of constitutively active receptors. Also highly analogous to 5-HT receptors in the ENS.
Line 46, page 2: “..has been shown to facility both…” I think is supposed to say “..has been shown to facilitate both….”
Line 50, page 2. Ref 14 is 10 years old. Insert updated reference here for ENS https://pubmed.ncbi.nlm.nih.gov/32152479/
Figures are clear and references formatted ok.
Author Response
Reviewer 1
This study investigates the possible role of group I metabotropic receptors in the enteric nervous system and potentially in colonic motility in isolated mouse colon. The results are clear and convincing. The data is novel and useful to the field. The appropriate time controls for bleaching calcium signals appears to have been performed. The immunohistochemistry seems sound and spatio-temporal mapping data of CMCs.
My major concern is the speculation that mGlu receptors are involved in normal ongoing colonic motility (CMCs), because the antagonists of mGlu receptors inhibited CMCs. This statement is not adequately supported by the data and needs revision. An alternative possibility is raised below and needs discussion, with some extra citations, for balance.
For example, line 445 states “In this study, we showed that Group I-mGlu receptors are involved in myenteric neurotransmission and for the first time, their involvement in colonic motility.” This is a strong overstatement and needs to be toned down.
RESPONSE: We thank the reviewer for their careful and constructive feedback. We have amended the line to read “In this study, we showed that Group-I mGlu receptors are involved in myenteric neurotransmission and that they may be involved in colonic motility.” (page 16).
There is no doubt the authors data clearly shows that mGlu receptor inhibition with AIDA reduces the calcium transients evoked by nerve stimulation. Again, the authors interpret this as evidence that glutamate is involved in neurotransmission. Another possibility is that glutamate is not involved in neurotransmission, but the antagonists reduce constitutively active glutamateric receptor activity in the soma of enteric nerve cell bodies and this drop in membrane resistance reduces neuronal excitability for Ach to generate fast cholinergic EPSPs.
As the authors appreciate, visceral organs like the gut express a variety of receptors. Some may have a clear functional role, but many have no known function (at present). The fact that mGlu receptors are present in the ENS has been known for some time. But, finding a synaptic input using intracellular electrophysiology that is mediated by glutamate has not been easy.
No mention is made (that I could see) of the possibility of constitutively active glutamatergic receptors. This definitely needs inclusion in the discussion.
RESPONSE: We thank the reviewer for raising this point and have added a paragraph to the Discussion (page 17) to address this.
It is noteworthy to remember that numerous 5-HT receptors exist in the ENS, but finding a functional role for endogenous 5-HT in synaptic transmission in mouse ENS has not been easy, even though antagonists for 5-HT3 receptors potently block CMCs (Spencer et al. 2013) https://pubmed.ncbi.nlm.nih.gov/23593931/ This should be mentioned and cited.
RESPONSE: We have included this information in the Discussion (page 17).
Line 550. “Our findings indicate that…. are involved in myenteric neuronal activity” is again too strong. Please reword “Our findings indicate that….. maybe involved in myenteric neuronal activity…”
RESPONSE: We have amended this sentence.
For example, in Figure 6 legend, it is stated “Group-I mGlu receptors are involved in colonic motility.” The data actually doesn’t fully support this. It certainly shows that the antagonists inhibit colonic motility. This is very different from implying that these receptors are involved in “normal” CMC activity.
RESPONSE: The figure legend has been amended to “Antagonists of Group-I mGlu receptors reduce colonic motility”.
The mGluR7 has been shown to exhibit constitutive activity. See: https://pubmed.ncbi.nlm.nih.gov/25881041/ This publication states “These findings introduce a novel potential physiological role for mGluR7 in the nervous system, that of a constitutively active receptor, and thereby suggest a model in which mGluR7 signaling may be impactful without the need to invoke strong receptor activation by millimolar concentrations of extracellular glutamate.” This reference should be quoted and mention made of the possibility of constitutively active receptors. Also highly analogous to 5-HT receptors in the ENS.
RESPONSE: We thank the reviewer for this suggestion. We have already addressed the possibility of constitutive activity for Group I mGlu receptors and have added a specific reference that shows constitutive activity of mGluR1/5. As the downstream mechanisms of mGluR7 are quite different (they belong to the Group III mGlu receptors) compared to the Group-I mGluRs, we believe it would not be relevant to include this.
Line 46, page 2: “..has been shown to facility both…” I think is supposed to say “..has been shown to facilitate both….”
RESPONSE: We have amended this sentence.
Line 50, page 2. Ref 14 is 10 years old. Insert updated reference here for ENS https://pubmed.ncbi.nlm.nih.gov/32152479/
RESPONSE: Thanks for the suggestion, we have added several new references to this sentence.
Reviewer 2 Report
Leembruggen et al. used Live calcium imaging and motility analysis on ex vivo preparations of mouse colon segments and found that mGlu5 played a role in modulating enteric neurotransmission but did not affect CMCs. Instead, mGlu1 regulated CMC frequency, but did not affect enteric neurotransmission. The study is well designed and the results support most of their conclusions. I have several concerns about this article:
1. Why did the authors use antagonist of mGluR1/5 to study the effect of these receptors instead of agonist?
2. The authors choose nNOS-positive neurons to study the effect of mGluR I group on enteric motor neurons. As we know that mGluR I receptors mainly participate in excitatory effect on neurons, why not choose Chat-positive neurons to study?
3. In page 531-4, Some of the changes above could also be attributed to the activity of enteric glial cells. Do glial cells respond to excitatory neurotransmitters with increased [Ca2+]i transients?
4. In page 509-10, Since it is possible that BAY 36-7620 influences CMC frequency by inhibiting mGlu1 activity on non-neuronal cells. Why not study the expression of mGluR 1 on pacemaker cajal cells of colon?
Author Response
Reviewer 2:
Leembruggen et al. used Live calcium imaging and motility analysis on ex vivo preparations of mouse colon segments and found that mGlu5 played a role in modulating enteric neurotransmission but did not affect CMCs. Instead, mGlu1 regulated CMC frequency, but did not affect enteric neurotransmission. The study is well designed and the results support most of their conclusions. I have several concerns about this article:
- Why did the authors use antagonist of mGluR1/5 to study the effect of these receptors instead of agonist?
RESPONSE: Both agonists and antagonists were added to investigate the role of mGluR1/5 neurotransmission using calcium imaging. In terms of the video imaging and analysis of colonic migrating motor complexes, we aimed to examine whether endogenous Group-I mGlu receptor activity was involved, therefore we applied antagonists to inhibit those receptors. Although the application of agonists would have been interesting to see the role of over-activation of these receptors on colonic motility, the continual exposure to an agonist for 1hr of video imaging time would have resulted in desensitisation of the receptors.
- The authors choose nNOS-positive neurons to study the effect of mGluR I group on enteric motor neurons. As we know that mGluR I receptors mainly participate in excitatory effect on neurons, why not choose Chat-positive neurons to study?
RESPONSE: We thank the reviewer for this suggestion. We chose to use 2 markers that were more specific for subtypes of enteric neurons: nNOS to label inhibitory motor neurons and calbindin to label intrinsic sensory neurons of the ENS. While acetyl choline is the major excitatory neurotransmitter, on the flip side, this means that many different subtypes of enteric neurons are ChAT-immunoreactive, including excitatory motor neurons, intrinsic sensory neurons as well as populations of interneurons (Qu et al., 2008 Cell Tiss Res). Therefore, we would not have been as effective at narrowing down what subpopulations are responding to mGluR1/5 inhibition. In addition, we would like to highlight that although mGluR1/5 activity is mostly excitatory, that does not prohibit it from acting on an inhibitory motor neuron; ie. The inhibitory (nNOS+) neuron is the postsynaptic recipient of the excitatory communication.
- In page 531-4, Some of the changes above could also be attributed to the activity of enteric glial cells. Do glial cells respond to excitatory neurotransmitters with increased [Ca2+]i transients?
RESPONSE: Previous studies have shown that enteric glial cells respond to several excitatory neurotransmitters, with increases in intracellular Ca2+ (Boesmans et al., 2013; Fung et al., 2017). We have added a sentence to the Discussion to include this information (page 17). In addition to our data showing that enteric glia are likely to respond to the Group-I mGlu receptor agonist DHPG (please see Figure 3, C&E). Given that our RNAscope data indicated the expression of Group I mGlu receptors in enteric glia (which has also been suggested by previous studies), it is likely that glia are involved in mGlu receptor activity in the myenteric plexus. How this influences neurotransmission and colonic motility would require further investigation, which we are interested in pursuing in the future.
- In page 509-10, Since it is possible that BAY 36-7620 influences CMC frequency by inhibiting mGlu1 activity on non-neuronal cells. Why not study the expression of mGluR 1 on pacemaker cajal cells of colon?
RESPONSE: We thank the reviewer for this useful suggestion. We wanted to understand the role of Group I mGluRs in myenteric neurons, and as such we made this the focus of our study. However, we agree that the role of mGlu receptors in non-neuronal cells needs to be considered and explored. We agree that investigating the function of Group I mGlu receptors (in particular mGluR1) in interstitial cells of Cajal would be a very interesting future experiment.